

# The utility of MEWS for predicting the mortality in the elderly adults with COVID-19: a retrospective cohort study with comparison to other predictive clinical scores

Lichun Wang[1,*], Qingquan Lv[2,*], Xiaofei Zhang[1], Binyan Jiang[3], Enhe Liu[4], Chaoxing Xiao[4], Xinyang Yu[3], Chunhua Yang[1] and Lei Chen[1]

[1] Department of Critical Care Medicine, The Sixth Affiliated Hospital of Sun Yat-Sen University, Guangzhou, Guangdong, China
[2] Department of Health Services Section, Wuhan Hankou Hospital, Wuhan, Hubei, China
[3] Department of Applied Mathematics, The Hong Kong Polytechnic University, Hongkong, China
[4] Department of Critical Care Medicine, Foresea Life Insurance Guangzhou General Hospital, Guangzhou, China
* These authors contributed equally to this work.

Corresponding author
Lei Chen, chenlei6@mail.sysu.edu.cn

## ABSTRACT

**Background:** Older adults have been reported to be a population with high-risk of death in the COVID-19 outbreak. Rapid detection of high-risk patients is crucial to reduce mortality in this population. The aim of this study was to evaluate the prognositc accuracy of the Modified Early Warning Score (MEWS) for in-hospital mortality in older adults with COVID-19.

**Methods:** A retrospective cohort study was conducted in Wuhan Hankou Hospital in China from 1 January 2020 to 29 February 2020. Receiver operating characteristic (ROC) analysis was used to evaluate the predictive value of MEWS, Acute Physiology and Chronic Health Evaluation II (APACHE II), Sequential Organ Function Assessment (SOFA), quick Sequential Organ Function Assessment (qSOFA), Pneumonia Severity Index (PSI), Combination of Confusion, Urea, Respiratory Rate, Blood Pressure, and Age ≥65 (CURB-65), and the Systemic Inflammatory Response Syndrome Criteria (SIRS) for in-hospital mortality. Logistic regression models were performed to detect the high-risk older adults with COVID-19.

**Results:** Among the 235 patients included in this study, 37 (15.74%) died and 131 (55.74%) were male, with an average age of 70.61 years (SD 8.02). ROC analysis suggested that the capacity of MEWS in predicting in-hospital mortality was as good as the APACHE II, SOFA, PSI and qSOFA (Difference in AUROC: MEWS vs. APACHE II, −0.025 (95% CI [−0.075 to 0.026]); MEWS vs. SOFA, −0.013 (95% CI [−0.049 to 0.024]); MEWS vs. PSI, −0.015 (95% CI [−0.065 to 0.035]); MEWS vs. qSOFA, 0.024 (95% CI [−0.029 to 0.076]), all $P > 0.05$), but was significantly higher than SIRS and CURB-65 (Difference in AUROC: MEWS vs. SIRS, 0.218 (95% CI [0.156–0.279]); MEWS vs. CURB-65, 0.064 (95% CI [0.002–0.125]), all $P < 0.05$). Logistic regression models implied that the male patients (≥75 years) had higher risk

of death than the other older adults (estimated coefficients: 1.16, $P = 0.044$). Our analysis further suggests that the cut-off points of the MEWS score for the male patients (≥75 years) subpopulation and the other elderly patients should be 2.5 and 3.5, respectively.

**Conclusions:** MEWS is an efficient tool for rapid assessment of elderly COVID-19 patients. MEWS has promising performance in predicting in-hospital mortality and identifying the high-risk group in elderly patients with COVID-19.

## INTRODUCTION

The WHO Director-General declared COVID-19 as a pandemic on 11 March 2020. According to the report of WHO-China Joint Mission on Coronavirus Disease 2019 (COVID-19) Report, (*Wang et al., 2019*) the older adults who are above 60 years old are the population with high-risk of death. The crude mortality of the older adults above 80 years old could be as high as 21.9%. An epidemiological investigation conducted by the Chinese Center for Disease Control and Prevention found that more than one-third of the COVID-19 patients were the older adults aged 60 years and above (*Wu & McGoogan, 2020*).

While older adults have been identified as the main group with COVID-19 infection, the vital signs of these patients could deteriorate very quickly. Hence, timely detection of the high-risk group and appropriately-assessed disease severity for the older adults with COVID-19 are practically essential (*Verity et al., 2020*). Several disease severity scoring systems are often used to guide the management of patients with pneumonia, such as the Acute Physiology and Chronic Health Evaluation II (APACHE II), Sequential Organ Function Assessment (SOFA), quick Sequential Organ Function Assessment (qSOFA), Pneumonia Severity Index (PSI), Combination of Confusion, Urea, Respiratory Rate, Blood Pressure, and Age ≥65 (CURB-65), Modified Early Warning Score (MEWS) and the Systemic Inflammatory Response Syndrome Criteria (SIRS)(*Asai et al., 2019*; *Chalmers et al., 2011*; *Dremsizov et al., 2006*; *Goodacre et al., 2010*; *Kim et al., 2019*; *Sikka et al., 2000*). All these scoring systems have been proven to be useful for predicting the outcome of patient's with pneumonia.

Among these scoring systems, MEWS can generally be obtained within minutes after the patient is admitted, providing a rapid assessment result for clinicians and enabling timely treatment to high risk patients (*Churpek et al., 2019*; *Salottolo et al., 2017*; *Yu et al., 2019*). Although MEWS has been found to be useful for assessment of pneumonia deterioration, (*Jo et al., 2016*), its utility has not been examined in predicting outcome of the older adults with COVID-19. The aim of this study is to evaluate the prognostic accuracy of MEWS for in-hospital mortality of older adults with with COVID-19.

## METHODS

### Study population

This is a retrospective cohort study conducted in the Wuhan Hankou Hospital in China from 1 January 2020 to 29 February 2020. It was approved by the Institutional Ethics Committee of Hankou hospital (hkyy2020-014). The Institutional Ethics Committee approved of waiving informed consents by reasons that this study was conducted retrospectively, privacy and personally identifiable information of the patients were protected and data collection would not harm the patient. The data in this study were analysed anonymously. Inclusion criteria were as follows: (1) Patients diagnosed as COVID-19 (Supplemental Text S1); (2) Patients treated by Guangdong medical team; (3) Older adults aged 60 or above. Exclusion criteria: (1) pneumonia caused by other pathogens; (2) Incomplete data.

### Clinical and biochemical data collection

All the data were obtained from the electronic medical records in the Information System of Wuhan Hankou Hospital. The information of the patients during admission including age, gender, underlying disease, consciousness state, vital signs, physiological and laboratory variables were collected and further used for the calculation of APACH II score, PSI score, SOFA score, qSOFA score and MEWS (Supplemental Text S2). All these variables/factors are objective measurable parameters collected by professional medical staff, and the laboratory variables were recorded and double check by two researchers.

### Outcomes variables

The main outcome of the study was in-hospital mortality. The other issue we tried to address was to identify the group of the high-risk patients based on MEWS. Patients were further categorised into different groups by their gender (male vs. female) and age (75 years old or above vs. 60–74 years old).

### Statistical analysis

Statistical analysis was carried out by Microsoft R (version 3.5.3). All the disease severity scores were described as Median (Inter-Quartile Range), while other numerical variables were described as Mean (SD). Categorical variables were reported as percentages and frequencies. Kolmogorov–Smirnov test was used to assess the normality of the numerical variables, and subsequently, we used $T$-test to compare normally distributed variables and used Mann-Whitney test to test whether the means between two groups are significantly different for non-normally distributed variables. Chi-square test was used to test independence for contingency tables of categorical variables. Receiver Operating Characteristic (ROC) curve was used to evaluate the predictive value of each scoring system for prognosis, and $Z$-test was used for the Area Under the ROC curve (AUROC). In order to evaluate the accuracy of each scoring system sensitivity, specificity, positive predictive value and negative predictive value were determined. Optimal cut-off value for each score was calculated using Youden's J-statistics. Bootstrapping confidence intervals were calculated to evaluate the difference in the AUROC between MEWS and the

other scores. Specifically, by resampling (with replacement) the data for 100 times, we obtained an empirical distribution for the difference of AUROC between MEWS and other scores. The confidence intervals were then obtained using the percentiles of the empirical distribution. In order to detect the high-risk population and understand the risk effect of the MEWS on the mortality rate in different subpopulations, logistic regression models were developed against MEWS, age groups, gender and the interaction between age group and gender. The optimal model was selected by Akaike Information Criterion (AIC). A smaller AIC score indicates a better fit of the corresponding model to the data. All the hypothesis tests were two-sided with a significance level of 0.05.

## RESULTS

### Study population and outcomes

The total number of patients treated by Guangdong medical team was 472, among which 244 (51.7%) met the enrolment criteria (patients aged 60 years old or above with a primary diagnosis of COVID-19 disease) from the 1 January 2020 to 29 February 2020. Nine patients were excluded from our study cause one patient had been diagnosed with pneumonia caused by other pathogens and eight patients had incomplete data. Therefore, there were 235 patients included in this analysis.

The general information and the clinical characteristics of the patients are listed in Table 1. The mean age was 70.6 years (SD 8.0) and 55.7% of them were male. A total of 43.8% of these patients had Hypertension, 26.4% of them had Diabetic Mellitus, and 20.9% had history of Coronary Heart Disease. The median values of scores in these patients were as follows: 12 (IQR: 9–17) in APACHE II score, 3 (IQR: 2–5) in SOFA score, 82 (IQR: 65–114) in PSI score, one (IQR: 1–2) in CURB-65, 2 (IQR: 1–4) in MEWS and 0 (IQR: 0–1) in qSOFA. 56.2% of these patients met SIRS criteria (Table 1).

Non-survivors were more likely to be male (27 (73.0%) vs. 104 (52.5%)), with higher APACHE II score (24 (IQR: 20–27) vs. 11 (IQR: 9–14)), SOFA score (7 (IQR: 6–9) vs. 3 (IQR: 2–4)), PSI score (152 (IQR: 128–166) vs. 75 (IQR: 63–100)), CURB-65 score (3 (IQR: 2–4) vs. 1 (IQR: 1–2)), MEWS score (5 (IQR: 4–6) vs. 2 (IQR: 1–3)) and qSOFA score (2 (IQR: 1–2) vs. 0 (IQR: 0–11)) than survivors. In addition, more non-survivors met SIRS criteria (33 (89.2%) vs. 99 (50.0%)).

Of the 235 patients during follow-up, 37 (15.8%) died in the hospital and the median duration of hospital stay of the non-survivors were 5.5 days (IQR: 2.0–9.3).

### Prognostic accuracy of MEWS in predicting the in-hospital mortality rate

The distributions of all the scores were presented at Fig. S1, and their relationships with in-hospital mortality in the older adults with COVID-19 were presented in Fig. 1. The AUROCs for these scores in predicting in-hospital mortality were as follows: APACHE II, 0.937 (95% CI [0.877–0.995]); SOFA, 0.926 (95% CI [0.877–0.975]); PSI, 0.927 (95% CI [0.898–0.986]); CURB-65, 0.845 (95% CI [0.740–0.951]); MEWS, 0.913 (95% CI [0.864–0.941]); SIRS, 0.696 (95% CI [0.616–0.776]); and qSOFA, 0.886 (95% CI [0.804–0.969]) (Table 2; Fig. 2).

**Table 1  Patient characteristics and outcome.**

|  | All patients | Survivors | Non-survivors | P |
|---|---|---|---|---|
| Number of patients | 235 | 198 | 37 | |
| Age, mean (SD), year | 70.6 (8.0) | 70.17 (9.9) | 72.95 (8.1) | 0.530 |
| Male, No. (%) | 131 (55.7) | 104 (52.5) | 27 (73.0) | 0.030 |
| Co-morbidity No. (%) | | | | |
| Hypertension | 103 (43.8) | 89 (45.0) | 14 (37.8) | 0.474 |
| Coronary heart disease | 49 (20.9) | 41 (20.7) | 8 (21.6) | >0.999 |
| Diabetes mellitus | 62 (26.4) | 52 (26.3) | 10 (27.0) | >0.999 |
| Chronic obstructive pulmonary disease | 31 (13.2) | 29 (14.7) | 3 (8.1) | 0.433 |
| Cerebrovascular disease | 19 (8.1) | 16 (8.0) | 3 (8.1) | >0.999 |
| Cancer | 8 (3.4) | 7 (3.5) | 1(2.7) | >0.999 |
| Others | 22 (9.4) | 21 (10.6) | 1 (2.7) | 0.215 |
| None | 67 (28.5) | 57 (28.8) | 12 (32.4) | >0.999 |
| Severity of Illness No. (%) | | | | |
| Mild | 98 (41.7) | 97 (49.0) | 1 (2.7) | <0.001 |
| Moderate | 48 (20.4) | 47 (23.7) | 1 (2.7) | 0.002 |
| Severe | 89 (37.9) | 54 (27.3) | 35 (94.6) | <0.001 |
| Scores on Admission, median (IQR), | | | | |
| APACHE II | 12 (9, 17) | 11 (9,14) | 24 (20, 27) | <0.001 |
| SOFA | 3 (2,5) | 3 (2,4) | 7 (6, 9) | <0.001 |
| PSI | 82 (65,114) | 75 (63,100) | 152 (128, 166) | <0.001 |
| CURB65 | 1 (1,2) | 1 (1,2) | 3 (2, 4) | <0.001 |
| MEWS | 2 (1, 4) | 2 (1,3) | 5 (4, 6) | <0.001 |
| qSOFA | 0 (0, 1) | 0 (0,1) | 2 (1,2) | <0.001 |
| SIRS criteria ≥2 No. (%) | 132 (56.2) | 99 (50) | 33 (89.19) | <.0001 |
| Outcome | | | | |
| Hospital mortality (primary outcome), No. (%) | 37 (15.8) | 0 | 37 (100) | <0.001 |
| Hospital length of stay, median (IQR), d | 13 (6,23) | 15 (8,24) | 5.5 (2.0,9.3) | <0.001 |

Note:
APACHE II, acute physiology and chronic health evaluation II; SOFA, sequential organ function assessment; qSOFA, quick sequential organ function assessment; PSI, pneumonia severity index; CURB-65, the combination of confusion, urea, respiratory rate, blood pressure, and Age ≥65; MEWS, modified early warning score; SIRS, systemic inflammatory response syndrome.

The reliability of MEWS in predicting in-hospital mortality was as good as the APACHE II score, SOFA score, PSI score and qSOFA (Difference in AUROC with MEWS: MEWS VS APACHE II, −0.025 (95% CI [−0.075 to 0.026], $P$ = 0.828)); MEWS vs. PSI, −0.013 (95% CI [−0.049 to 0.024], $P$ = 0.748); MEWS vs. PSI, −0.015 (95% CI [−0.065 to 0.035], $P$ = 0.735); MEWS vs. qSOFA, 0.024 (95% CI [−0.029 to 0.076], $P$ = 0.174), but the prognostic accuracy of MEWS was significantly higher as compared with either SIRS criteria or CURB-65 score (Difference in AUROC with MEWS: MEWS vs. SIRS, 0.218 (95% CI [0.156–0.279], $P$ < 0.001); MEWS vs. CURB-65, 0.064 (95% CI [0.002–0.125], $P$ = 0.015)). The optimal cut-off value of MEWS for predicting the inpatient mortality in the older adults with COVID-19 was 4.5, with a specificity of 94.5% and a sensitivity of 67.6%.

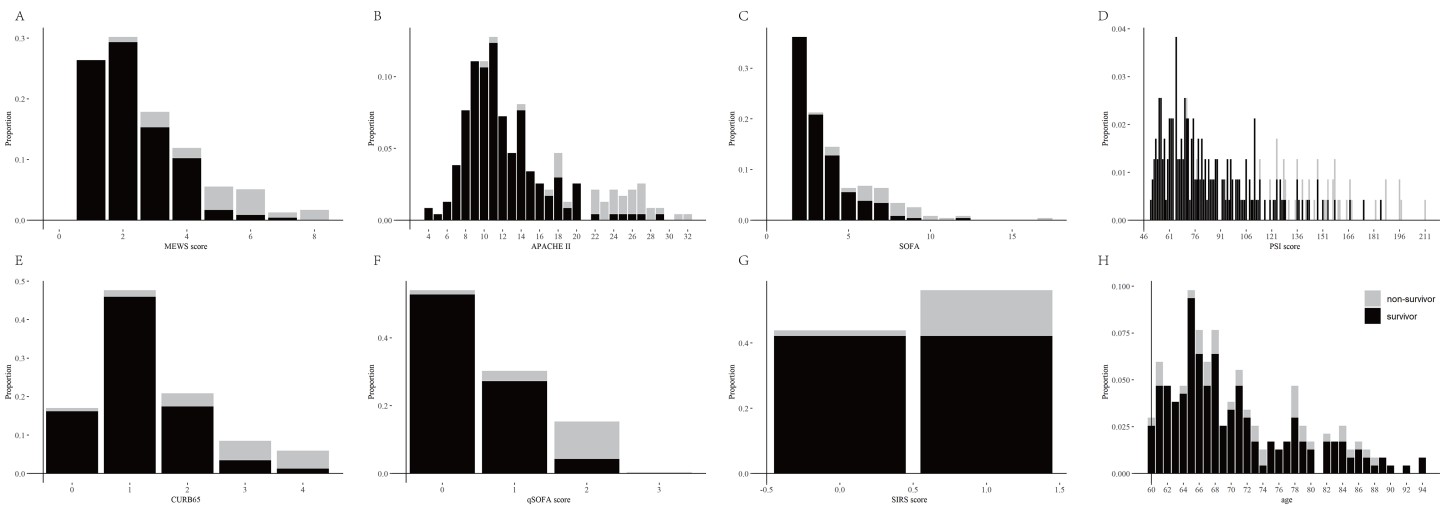

**Figure 1 The proportion of survivors and non-survivors in older adults with COVID-19 by MEWS, APACHE II, SOFA, PSI, CURB-65, qSOFA, SIRS and age.** The proportion of survivors and Non-Survivors in older adults with COVID-19 by MEWS (A), APACHE II (B), SOFA (C), PSI (D), CURB-65 (E), qSOFA (F), SIRS (G) and age (H).

**Table 2 Prognostic accuracy of difference score in predicting the in-hospital mortality and the difference with AUROC of MEWS.**

| | AUROC (95% CI) | Age-adjusted AUROC (95% CI) | Cut-off | SEN (%) | SPE (%) | NNPV (%) | PPV (%) | Difference with AUROC of MEWS (95% CI) | P |
|---|---|---|---|---|---|---|---|---|---|
| APACHE II | 0.937 [0.877–0.995] | 0.0943 [0.886–0.984] | 16.5 | 91.9 | 87.4 | 98.3 | 57.6 | −0.025 [−0.075 to 0.026] | 0.828 |
| SOFA | 0.926 [0.877–0.975] | 0.927 [0.87–0.969] | 5.5 | 81.1 | 89.4 | 96.2 | 58.8 | −0.013 [−0.049 to 0.024] | 0.748 |
| PSI | 0.927 [0.898–0.986] | 0.934 [0.879–0.976] | 113.5 | 91.9 | 86.4 | 98.3 | 55.7 | −0.015 [−0.065 to 0.035] | 0.735 |
| CURB-65 | 0.845 [0.740–0.951] | 0.833 [0.739–0.912] | 1.5 | 83.8 | 73.7 | 96.1 | 37.4 | 0.064 [0.002–0.125] | 0.015 |
| MEWS | 0.913 [0.864–0.941] | 0.913 [0.853–0.960] | 4.5 | 67.6 | 94.5 | 94.1 | 78.1 | — | |
| SIRS criteria | 0.696 [0.616–0.776] | 0.776 [0.700–0.842] | 0.5 | 89.9 | 50.0 | 96.1 | 25.0 | 0.218 [0.156–0.279] | <0.001 |
| qSOFA | 0.886 [0.804–0.969] | 0.899 [0.818–0.956] | 1.5 | 73.0 | 95.0 | 95.0 | 73.0 | 0.024 [−0.029 to 0.076] | 0.174 |

**Note:**
APACHE II, acute physiology and chronic health evaluation II; SOFA, sequential organ function assessment; qSOFA, quick sequential organ function assessment; PSI, pneumonia severity index; CURB-65, the combination of confusion, urea, respiratory rate, blood pressure, and Age ≥65; MEWS, modified early warning score; SIRS, systemic inflammatory response syndrome; AUROC, area under the receiver operating characteristic curve; SEN, sensitivity; SPE, specificity; NPV, negative predictive value; PPV, positive predictive value.

We also analysed the age adjusted ROC curves and AUROCs to evaluate the prognostic accuracy of MEWS. However, no significant difference was observed from the original analysis without age adjustment (Table 1).

## Identifying the risk population with high mortality via logistic regression models

In order to identify the high-risk older adults with COVID-19 and understand the prognostic accuracy of the MEWS on the mortality in different subpopulations categorised by gender and age group, logistic regression models were developed. In particular, an interaction term age × gender term was introduced to consider possible interaction effect between age group and gender. The optimal model was selected by AIC. The best model is the one with features age × gender (estimated coefficients: 1.16, P = 0.044) + MEWS

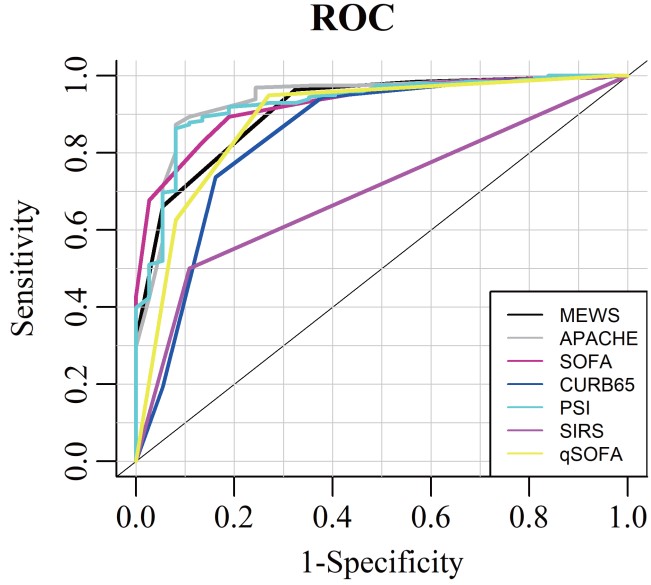

**ROC**

**Figure 2 Area under the receiver operating characteristic curve to discriminate in-hospital mortality for MEWS, APACHE II, SOFA, PSI, CURB-65, SISR and qSOFA.** ROC analysis suggested that the capacity of MEWS in predicting in-hospital mortality was as good as the APACHE II, SOFA, PSI and qSOFA, but was significantly higher than SIRS and CURB-65.

**Table 3 Logistic regression models of the age group, gender and MEWS.**

| Model | AIC | Estimated coefficients | | | | Standard error | | | | P | | | |
|---|---|---|---|---|---|---|---|---|---|---|---|---|---|
| | | Age | Gender | Gender*Age | MEWS | Age | Gender | Gender*Age | MEWS | Age | Gender | Gender*Age | MEWS |
| 1 | 118.01 | – | – | – | 1.24 | – | – | – | 0.18 | – | – | - | <0.001 |
| 2 | 116.03 | – | – | 1.16 | 1.24 | – | – | 0.58 | 0.18 | – | – | 0.044 | <0.001 |
| 3 | 117.68 | 0.79 | – | – | 1.23 | 0.51 | – | – | 0.18 | 0.125 | – | – | <0.001 |
| 4 | 116.54 | – | 0.96 | – | 1.24 | – | 0.53 | – | 0.19 | – | 0.071 | – | <0.001 |
| 5 | 118.03 | 0.04 | – | 1.13 | 1.24 | 0.82 | – | 0.91 | 0.18 | 0.964 | – | 0.214 | <0.001 |
| 6 | 118.52 | 0.50 | 0.80 | 0.34 | 1.24 | 0.92 | 0.66 | 1.12 | 0.19 | 0.584 | 0.228 | 0.764 | <0.001 |
| 7 | 116.61 | 0.73 | 0.91 | – | 1.24 | 0.52 | 0.54 | – | 0.19 | 0.163 | 0.089 | – | <0.001 |
| 8 | 116.81 | – | 0.65 | 0.84 | 1.24 | – | 0.59 | 0.64 | 0.19 | – | 0.272 | 0.189 | <0.001 |

**Note:**
MEWS, modified early warning score. Model 1: MEWS; Model 2: Gender*Age + MEWS; Model 3: Age + MEWS; Model 4: Gender + eMEWS; Model 5: Age + Gender*Age + MEWS; Model 6: Age + Gender*Age + MEWS + Gender; Model 7: Age + Gender + MEWS; Model 8: Gender*Age + Gender + MEWS. Logistic regression models were developed against MEWS, age groups ('1' for 75 years old or above, and '0' for 60–74 years old), gender ('1' for male and '0' for female), and the interaction between age group and gender (age*gender).

(estimated coefficients: 1.24, $P < 0.001$) (Table 3). Note that under our notations, the age × gender indicator corresponds to whether the patient is a male patient aged 75 years old or above. The positive coefficient (1.16) thus implies the group of COVID-19 patients who were male and aged 75 years or above had higher risk of death than the other COVID-19 patients in this population. In particular, since the coefficient of age × gender (1.16) is very close to the coefficient of the MEWS (1.24), that is, 1.16/1.24 ≈ 1, the risk of male COVID-19 patients aged 75 years and above for a given MEWS is as high as

the risk of other patients with a one mark higher MEWS score. Motivated by this, we further explore the risk effect of the MEWS in the two subpopulations (male patients aged 75 years or above VS other patients in this population) separately. Results on the prognostic accuracy of MEWS in predicting the in-hospital mortality of these two subpopulations were showed in Table S1 and Fig. S2. The optimal cut-off value of MEWS in the male COVID-19 patients aged 75 years or above was 2.5, with 84.3% specificity, 84.6% sensitivity, 57.9% mortality rate for those with MEWS > 2.5 and 0.0% mortality rate for those with MEWS < 2.5. While for other older adults with COVID-19, the optimal cut-off value of MEWS was 3.5, with 75.0% specificity, 100% sensitivity, 45.8% mortality rate for those with MEWS > 3.5 and 2.8% mortality rate for those with MEWS < 3.5. We remark that there is a one point difference in the cut-off points of the two subpopulations (2.5 vs. 3.5). This is consistent to the results we have discussed for the fitted model.

## DISCUSSION

In this retrospective cohort study, ROC analysis suggested that prognostic accuracy of MEWS in predicting the inpatient mortality rate is comparable to that of APACHE II score, PSI score, SOFA score, and better than the CURB-65, SIRS criteria in older adults with COVID-19. On the other hand, comparing to APACHE II, PSI and SOFA, MEWS is much simpler and can be rapidly obtained in clinical practice. Our analysis has also shown MEWS to be a promising tool for risk stratification of these patients. We thus conclude that MEWS is an effective prognostic tool for providing rapid assessment and identifying the high-risk patients among the older adults with COVID-19, so the use of MEWS should be encouraged in this population during this severe plague outbreak.

Since the outbreak of COVID-19 has begun, as of July 103, 2020, more than 12 million cases over the world have been confirmed and over half million deaths have occurred. COVID-19 has resulted in considerable morbidity and mortality in more than 200 countries worldwide. There is no specific drug for treating the patients with COVID-19, till mow treatment is mainly symptomatic. Recent literatures have identified older adults group are more prone towards COVID-19 infection, and they might be at higher risk of death than younger patients (*Liu et al., 2020*; *Onder, Rezza & Brusaferro, 2020*; *Verity et al., 2020*; *Wu & McGoogan, 2020*) It is reported that the mortality in the patients aged 60 years or above ranged from 5.98% to 12.4%, while the mortality in the younger patients ranged from 1.32% to 0.64% (*Onder, Rezza & Brusaferro, 2020*; *Wu & McGoogan, 2020*). In our study, mortality in older adults with COVID-19 aged 60 years or above is 15.8%. A recent study demonstrated that older adults aged 65 years or above with COVID-19 are more likely to develop organ injury in a short time, leading to an increased risk of death (*Yang et al., 2020*). In our study, we found that the median duration of hospital stay of the non-survivors was 5.5 days (IQR: 2.0–9.3), indicating that the vital signs of these patients could deteriorate very quickly. Early identification of the high-risk patients and timely medical interventions could possibly reduce the high mortality rate among this population. Rapid and effective assessment of these older adults with

COVID-19 is thus very crucial. Owning to the rapid spread of COVID-19, there has been a severe shortage of medical resources, so rapid assessment could potentially serve as an essential tool for efficient allocation of these limited resources to those who are indeed at has been high risk of death.

Pneumonia Severity Index score and CURB-65 score have been wildly used for prognostic evaluations and severity assessment of pneumonia, which enables the clinicians to allocate the patients to appropriate level and intensity of care and perform effective medical management in time (*Kim et al., 2019*; *Ning et al., 2018*). It had reported that high PSI score, CURB-65 score, APACHE II score and SOFA score were associated with poor outcome in the patients with pneumonia (*Asai et al., 2019*; *Kao et al., 2018*). Since the outbreak of COVID-19, APACHE II score, CURB-65 score, SOFA score and qSOFA score have been used to assess the severity of COVID-19 patients (*Wang et al., 2020*; *Yang et al., 2020*; *Zhou et al., 2020*). In a cohort of 52 patients with COVID 19, non-survivors had higher APACHE II score than the survivors (18 (IQR: 16–20) vs. 14 (IQR : 12–17)) (*Yang et al., 2020*). A recent study conducted by *Zhou et al. (2020)* indicated that non-survivors had higher SOFA score (4.5 (IQR: 4.0–6.0) vs. 1.0 (IQR: 1.0–2.0), $P < 0.0001$), qSOFA score (1 (IQR: 1.0–2.0) vs. 0 (IQR: 0.00–1.0), $P < 0.0001$), CURB-65 (2.0 (IQR: 1.0–3.0) vs. 1.0 (IQR: 0.0–1.0), $P < 0.0001$) than survivors. It is worth mentioning that, non-survivors in the older adult COVID-19 population we studied had higher APACHE II score (24 (IQR: 20–27)), SOFA score (7 (IQR: 6.0–9.0)), CURB-65 score (3 (IQR: 2.0–4.0)) and qSOFA score (2 (IQR: 1.0–2.0)), compared to previous studies with different populations. Although APACHE II, SOFA, PSI and CURB-65 work well in clinical practice, they generally require sophisticated information from the patient and are not suitable for rapid assessment.

Modified Early Warning Score consists of 5 physiological parameters, which include systolic blood pressure, pulse rate, respiratory rate, temperature, and level of consciousness. It has been used as an essential tool for early identification of patients who can deteriorate in medical and surgical wards (*Fullerton et al., 2012*; *Kramer, Sebat & Lissauer, 2019*; *Salottolo et al., 2017*). Some studies have shown that a MEWS score of 5 or above is associated with a high risk of poor outcome in intensive care unit (*Reini, Fredrikson & Oscarsson, 2012*). Comparing with other scores, MEWS is much simpler and more importantly allows rapid assessment and dynamic monitoring. However, the capacity of MEWS on COVID-19 patients hasn't been well explored yet. This retrospective cohort study focused on the utility of MEWS in predicting outcome in the older adults with COVID-19. We observed that, for the older adults with COVID-19, the capacity of MEWS to predict in-hospital mortality was significantly higher as compared with SIRS criteria and CURB-65 score, while the prognostic accuracy is comparable to those of APACHE II, PSI and SOFA. More importantly, MEWS allows early detection of the high-risk older adults with COVID-19. We found that the cut-off values using MEWS to predict the outcome in COVID-19 is smaller than the five points in the guidelines for other disease. In addition, the present study reveals that the male COVID-19 patients aged 75 years or above had higher risk of death than the other older adults with COVID-19. In particular, the cut-off points of the MEWS for the male COVID-19 patients

aged 75 years or above should be one-point lower than the cut-off for the other older adults with COVID-19 (2.5 vs. 3.5). It is worth mentioning that when the male patients aged 75 years or above were compared to the rest of the older adults, the true positive rate of MEWS's cut-off points can be as high as 100% for the male aged 75 years or above group, and 84.62% for the other group. This indicates that with our proposed cut-off scores, MEWS works very well in correctly identifying the true high-risk patients. We suggest that these cut-off points could potentially be used as a simple reference to rapidly identify the high-risk older adults in the near future.

## LIMITATION

There are some limitations in this study. Firstly, it was a retrospective study conducted in a single centre with relatively small sample size. Secondly, this study did not examine the younger population, but it may enable us to generalise the results to the other population. Thirdly, it would be interesting to explore how the heterogeneity in treatments over time would affects the effectiveness of MEWS and other scores. Statistically speaking, if such kind of data is available, more sophisticated models (e.g. logistic regression with mixed effects, or varying coefficient models) can be conducted to take into account possible heterogeneity in the data, and obtain better understandings for the utility of MEWS. Last but not least, although this study revealed that there is a significant association between MEWS and mortality in elderly patients with COVID-19 upon admission, questions such as whether or not subsequent treatments are associated with the reduction of MEWS, and whether or not the reduction of MEWS in subsequent treatments is associated with lower mortality, are yet to be further investigated in future studies.

## CONCLUSION

Our retrospective cohort study showed that, in terms of prognostic accuracy, MEWS performed as good as APACHE II, PSI and SOFA, and outperformed CURB-65 and SIRS. MEWS has proven to be a promising tool for risk stratifications among the older adults COVID-19 population in this study. We have not only found MEWS to be an important variable for the risk, but also identified the subpopulation consists of male COVID-19 patients aged 75 years or above as the group with higher risk of mortality. Our analysis further suggested a cut-off point of MEWS score 2.5 for male COVID-19 patients aged 75 years or above and a cut-off point of MEWS score 3.5 for other older adults with COVID-19.

### Funding

The authors received no funding for this work.

### Competing Interests

The authors declare that they have no competing interests.

## Author Contributions

- Lichun Wang conceived and designed the experiments, performed the experiments, analysed the data, prepared figures and/or tables, authored or reviewed drafts of the paper, and approved the final draft.
- Qingquan Lv performed the experiments, authored or reviewed drafts of the paper, and approved the final draft.
- Xiaofei Zhang conceived and designed the experiments, authored or reviewed drafts of the paper, and approved the final draft.
- Binyan Jiang analysed the data, prepared figures and/or tables, and approved the final draft.
- Enhe Liu performed the experiments, analysed the data, authored or reviewed drafts of the paper, materials, and approved the final draft.
- Chaoxing Xiao performed the experiments, analysed the data, prepared figures and/or tables, authored or reviewed drafts of the paper, analysis tools, and approved the final draft.
- Xinyang Yu analysed the data, prepared figures and/or tables, and approved the final draft.
- Chunhua Yang performed the experiments, authored or reviewed drafts of the paper, and approved the final draft.
- Lei Chen conceived and designed the experiments, authored or reviewed drafts of the paper, and approved the final draft.

## Human Ethics

The following information was supplied relating to ethical approvals (i.e., approving body and any reference numbers):

This study was approved by the Institutional Ethics Committee of Hankou hospital (Ethical Application Ref: hkyy2020-014).

## Data Availability

The raw measurements are available in the Supplemental Files.

## Supplemental Information

Supplemental information for this article can be found online at http://dx.doi.org/10.7717/peerj.10018#supplemental-information.

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
