# Peer review of "The utility of MEWS for predicting the mortality in the elderly adults with COVID-19: a retrospective cohort study with comparison to other predictive clinical scores"

_PeerJ, doi:10.7717/peerj.10018_

## Round 0.1 · original submission · Major Revisions

The three reviewers have somewhat conflicting views on your manuscript. Regardless of these differences, I feel you need to take on board these points in order to prepare a resubmission of this manuscript. In particular, think about any of the confounding variables that may have influenced the assessment of exposure as well as the wide variety of outcome data that was collected. In particular, please provide more details on how all of this data was accurately obtained and recorded.

Reviewer 1 ·

Basic reporting

no comment

Experimental design

1- The research is original
2- research question is well defined
3 -The investigation is not so rigorous, since the study design does not allow it
4- Methods are described with sufficient detail

Validity of the findings

1- I don´t see any novelty or impact resulting from the study
2- All the data provided seems to be ok
3- I cannot relate what they did with the conclusion made.
4- My criticism is about the study design and the many instruments used to predict outcome. Since we cannot control exposure or outcome assessment, we must rely on others for accurate recordkeeping.

Additional comments

- Despite of being well written, the research was not planned to answer this research question. It seems like the data comes from an umbrela study and this is a piece. Thus, we might mislead with this research. - I do not see a novelty approach in this.

Reviewer 2 ·

Basic reporting

The English language should be improved to ensure that an international audience can clearly understand your text. I suggest that a native English speaking colleague review your manuscript. The current version makes comprehension difficult.

Experimental design

Some details can be specified in the "Statistical Analysis" Section.
There is an informed consent but only in Chinese.

Validity of the findings

No comment

Additional comments

The Authors have evaluated the prognostic accuracy of Modified Early Warning Score (MEWS) in order to estimate the in-hospital mortality, in older adults with COVID-19. The retrospective cohort study was conducted in Wuhan Hankou Hospital from 1st January to 29th February 2020.
The topic is well developed and potentially interesting for readers. However, I have some minor criticisms:
- In the "Statistical Analysis" Section, it would be indicated: i) when the data were summarized with the means and when with the medians; ii) if p values were adjusted for multiple comparisons; iii) how the differences between AUROCS of different scores vs AUROC of MEWS were estimated (see also Table 2);
- Informed consent is only available in Chinese;
- Some typos and / or repetitions must be corrected (for example: line 77, 85, 270, 271, etc.);
- Sometimes punctuation is missing (for example: pag. 16, 17, etc.);
- Some sentences are not clear and need to be reworded (for example: line 83, 84, 162, 163, 274, 275, etc.);
- There is no concordance between the times in the present and in the past, or between singular and plural (for example: line 146, 147, etc.);
- The text must be corrected by an English native speaker, otherwise it is difficult to read.

·

Basic reporting

The title needs to be revised. The manuscript did not only check MEWS but it also checked other scoring systems and made comparisons. Please revise the Title to mention the comparison of different scoring systems.

Experimental design

". A waiver for informed consent was provided for this retrospective cohort study" looked strange. There is no such waiver of informed consent. Was the study reviewed and approved by the IRB? Not mentioned in the Methods. It should be "IRB approved the waiving of informed consents...". A waiver could not be provided. By whom? To whom? Please revise it.

Validity of the findings

In the paragraph of statistical methods, there was no need to specify the coding or labeling of the nominal variables such as gender or ordinal variables such as age groups. There was also no need to specify the R packages. Please ask a qualified statistician to revise the whole paragraph.

Additional comments

1. Please revise the Title to mention the comparison of different scoring systems.
2. Please mention IRB in the Methods.
3. Please revise the paragraph of statistical methods with the advice from statisticians.

---

## Round 0.2 · Minor Revisions

I sincerely thank the authors for attending to the majority of comments from three reviewers on the first version of the submitted manuscript. There are still some areas identified by the reviewers that need to be addressed before this manuscript can be more seriously considered for publication.

Reviewer 2 ·

Basic reporting

Please, correct "INTRODUCTION" (row 69)

Experimental design

In the "Statistical analysis" Section, explain whether the normality of the data distribution has been assessed. Generally mean and SD are used for continuous variables normally distributed and median and interquartile range for non continuous distributions. Please, write it explicitly in the Section.

Validity of the findings

No comment

Additional comments

In the "Statistical analysis" Section, explain whether the normality of the data distribution has been assessed. Generally mean and SD are used for continuous variables normally distributed and median and interquartile range for non continuous distributions. Please, write it explicitly in the Section.

·

Basic reporting

1. Please revise the Title to make it concise and informative. Please specify "mortality" rather than "outcome". There is no need to list all other scores. Just write "...other predictive clinical scores...".

Experimental design

1. For the inclusion criteria, the old COVID-19 patients in Wuhan Hankou Hospital were included in the study. But why did they have the inclusion criteria (2): treated by Guangdong medical team? Why the patients in Wuhan were treated by Guangdong medical team? Please explain. Why not included the patients in Wuhan treated by non-Guangdong medical team?

Validity of the findings

1. If the patients' MEWS improved, would they have lowered mortality afterwards? The point is, did the initial MEWS determine the mortality, regardless any subsequent treatments? Or, the MEWS reflected the clinical severity? We could improved the MEWS to reduce the morality?

Additional comments

1. Please revise the Title to make it concise and informative. Please specify "mortality" rather than "outcome". There is no need to list all other scores. Just write "...other predictive clinical scores...".
2. For the inclusion criteria, the old COVID-19 patients in Wuhan Hankou Hospital were included in the study. But why did they have the inclusion criteria (2): treated by Guangdong medical team? Why the patients in Wuhan were treated by Guangdong medical team? Please explain. Why not included the patients in Wuhan treated by non-Guangdong medical team?
3. If the patients' MEWS improved, would they have lowered mortality afterwards? The point is, did the initial MEWS determine the mortality, regardless any subsequent treatments? Or, the MEWS reflected the clinical severity? We could improved the MEWS to reduce the morality?

---

## Round 0.3 · accepted · Accept

Based on the comments of the two reviewers, I am happy to recommend this paper be accepted for publication in PeerJ.

Reviewer 2 ·

Basic reporting

No comment

Experimental design

No comment

Validity of the findings

No comment

Additional comments

I have no further comments

·

Basic reporting

Good.

Experimental design

Good.

Validity of the findings

Good.

Additional comments

My concerns are addressed. Thanks.